# The Preparations of Fluorographene Nanosheets and Research in Tribological Properties in High Vacuum

**DOI:** 10.3390/ma16113929

**Published:** 2023-05-24

**Authors:** Lili Zhang, Zhengrui Zhang, Xi’an Gao, Siti Jahara Matlan, Nazaruddin Abd Taha

**Affiliations:** 1College of Civil Engineering, Sichuan University of Science & Engineering, Zigong 643000, China; zhanglili@suse.edu.cn (L.Z.); gaoxi_an@163.com (X.G.); 2Civil Engineering Programme, Faculty of Engineering, University of Malaysia Sabah, Kota Kinabalu 88400, Malaysia; jahara@ums.edu.my (S.J.M.); nazardin@ums.edu.my (N.A.T.)

**Keywords:** FG nanosheets, solvent-ultrasonic exfoliation, tribological properties, preparation, characterization

## Abstract

In this study, fluorographene nanosheets (FG nanosheets) were prepared via the solvent-ultrasonic exfoliation method. The fluorographene sheets were observed using field-emission scanning electron microscopy (FE-SEM). The microstructure of the as-prepared FG nanosheets was characterized by X-ray diffraction (XRD) and a thermal analyzer (TG). The tribological properties of FG nanosheets as an additive in ionic liquids in high vacuum were compared to that of ionic liquid (IL) with graphene (IL-G). The wear surfaces and transfer films were analyzed via an optical microscope, Raman spectroscopy, scanning electron microscopy (SEM), and X-ray photoelectron spectroscopy (XPS). The results show that FG nanosheets can be obtained from the simple solvent-ultrasonic exfoliation method. The prepared G nanosheets are a sheet, and the longer the ultrasonic time is, the thinner the sheet is. Ionic liquids with FG nanosheets had low friction and a low wear rate under high vacuum conditions. The improved frictional properties were attributed to the transfer film of FG nanosheets and more formation film of Fe-F.

## 1. Introduction

The development of space science puts forward high requirements for the long life and high reliability of space moving parts mechanisms. Good lubrication technology is one of the key technologies to improve the long life of space moving parts. Liquid lubrication has a high bearing capacity, good heat transfer performance, and good high-speed lubrication performance, but in high temperatures it will not form a good lubricating oil film due to reduced viscosity, and at low temperatures it will lose fluidity due to excessive viscosity, reducing the lubrication’s performance. Low speeds impact the boundary lubrication condition, making it difficult to form a continuous lubrication film, leading to direct contact between two local metal surfaces, causing serious wear, friction heat, and grinding. This will affect the chemical stability of the lubricating oil, leading to early lubrication failure. The design of the structure of liquid lubrication is relatively complex as it is a heavyweight liquid. In order to meet the needs of high reliability and long life of space moving parts, exploring and developing new lubrication materials and technologies is necessary.

It has been found that adding nanosolid particles to liquid lubricant can not only improve the film-forming performance [1,2] but also fill the micropits and damage sites on the workpiece surface, which may be realized the in situ repair of the friction surface. In recent years, researchers have carried out a series of works on nanoparticles as lubricating oil additives, including the tribological properties, mechanism of tribology, and the preparation and postprocessing of nanoparticles used in this aspect. At present, the studied nanoparticles used as lubricating oil additives are mainly nanopowder, nanoscale oxide [3,4], nanohydroxide, nanosulfide [5,6,7], nanoborate, polymer nanospheres [8], nanorare earth compounds, and low dimensional carbon-based nano additives, among others [9,10,11].

Graphene in carbon-based nano additives is a two-dimensional thin film composed of a hexagonal honeycomb lattice with an sp and sp^2^ hybrid orbit, with a thickness of only one carbon atom. Graphene has excellent physical properties, including mechanical strength, tensile stress, thermal conductivity, etc., and these properties make graphene an appropriate additive, namely one that is durable, has heat resistance, and has a low friction coefficient. In the field of lubrication additives, it is found that graphene can form an early transfer film in the friction process, avoiding the direct contact of the friction pair, thus playing the role of reducing friction and grinding [9,10]. He Yongyong’s group compared the tribological behavior of graphene as a lubricant additive for steel/copper and steel/steel friction pairs [12]. For the steel/copper friction pairs, the graphene sheet significantly reduced the friction coefficient and wearing depth at low load but then slightly increased at high load. The steel/steel friction pair shows excellent tribological properties even at high loads. Severe plastic deformation on the copper surface will reduce the stability of the graphene friction film because a rough copper transfer film forms on the steel surface during the run-in. In addition, Mn_3_O_4_ nanoparticles and graphene were recombined to obtain Mn_3_O_4_@G [11], and Mn_3_O_4_ nanoparticles were anchored in the graphene surface and sheet layer. The result has excellent tribological properties and high stability due to the synergistic lubrication between the graphene nanosheets and Mn_3_O_4_ nanoparticles. Even at ultralow concentrations (0.075 wt%) and 125 °C, the friction coefficient and wear depth were reduced by 75% and 97%, respectively, compared to the base oil. The synthetic methods and Mn_3_O_4_ @G nanocomposites have significant potential for energy saving in various tribological applications.

Jiang et al. studied the tribological behavior of graphene and graphene oxide (GO) as water-based lubricant additives [13]. The results indicate that the tribological behavior of water can be improved by adding appropriate graphene or GO. 0.5 wt% of graphene reduced the friction coefficient by 21.9% and the wear rate by 13.5% compared with pure DI water. Meanwhile, 0.5 wt% GO nanofluids were found to reduce the friction coefficient and wear rate to 77.5% and 90%, respectively. Zhang et al. successfully grafted amino-terminated block copolymer to the graphene surface by surface sulfonation and neutralization [14], and the modified graphene showed macroscopic fluidity in the absence of solvents at room temperature. The excellent solubility of liquid-like graphene enables good dispersion of graphene in a wide spectrum of solvents for a long time. As a lubricant additive in water, the maximum reduction of the friction coefficient and wear rate can reach 53% and 91%, respectively, compared with water, which may be due to the formation of tribo-film on the contact interface by electrostatic adsorption and molecular rearrangement of the liquid-like graphene. Cao et al. prepared a silver/graphene nanocomposite [15] by laser irradiation, which effectively avoided aggregation and had good dispersion stability. Monodispersed silver (Ag) nanospheres grow uniformly on layered graphene sheets, and this regular layered structure further ensures enhanced lubrication. Tribology experiments showed that the addition of 0.1 wt% of this composite reduced the friction coefficient and the wear diameter by 40% and 36%, respectively. Feng’s team built a sustainable lubrication system [16] by generating graphene in situ, thus avoiding the complex graphene synthesis process. A new lubrication system was constructed by introducing cyclopropane carboxylic acid (CPCa) and metal particles (nickel and copper) into the base oil. CPCa molecules undergo frictional chemical reactions to form hydrocarbons as solid lubricants and metal particles catalyze the graphitization of hydrocarbons. Importantly, friction products can be reused as lubrication additives for base oil and have better lubrication performance compared to the original lubrication system.

The derivatives of graphene mainly include graphene oxide, graphane, and fluorinated graphene (FG) [17,18,19,20,21,22]. Fluorinated graphene includes partially fluorinated graphene and all fluorinated graphene. It not only maintains the high strength of graphene but also nearly doubles the interlayer spacing of carbon atoms in the molecular structure due to the introduction of fluorine atoms. As a result, the interlayer force is weakened, and the repulsion between fluorine atoms is also conducive to interlayer sliding, so its lubrication performance is significantly better than graphene [23]. If fluorinated graphene is used instead of graphene, the performance of lubricating oil can be further improved. In recent years, researchers have explored the tribological performance [24] of fluorinated graphene in PAO, soybean oil, GTL, water, and other systems, and the results show that it can significantly improve the antifriction performance of fluid lubricants. Zheng et al. prepared a less-fluorinated graphene sheet [25] by the method of ball grinding. Compared with the fluorinated graphene sheet, it had better dispersion stability in the base oil PAO-8, with an optimal addition concentration of 0.30 mg/mL. Zhao et al. believed that the higher the fluorination degree of fluorinated graphene is [26], the more tribological performance will be further improved. Wang’s team prepared the hydrophilic urea-modified FG to improve the dispersion [27] of fluorinated graphene in water. Professor Liu grafted acrylic acid (AA) to the surface of FG [28], which improved the dispersion of FG in water. As a water-based lubricant additive, FG-AA shows excellent tribological properties. Compared with FG-PEI, its friction coefficient and wear rate are reduced by about 66% and 82%, respectively.

At present, the preparation of fluorinated graphene mainly includes the following methods [29,30,31,32,33,34]: One is graphene as a raw material with fluorinated reagents to obtain fluorinated graphene. However, the raw graphene material is difficult to prepare and expensive, and the fluorination reagent is not easy to obtain and is toxic, so the reaction has high requirements for equipment. Another method is to strip the fluorinated graphite for the fluorinated graphene [35,36,37,38,39]. There are mechanical and liquid phase exfoliations, and the liquid phase exfoliation uses organic solvent [40] or ionic liquid [41,42]. Liquid phase ultrasonic exfoliation has the advantages of simple operation and low energy consumption. Therefore, this paper uses fluorinated graphite as the raw material to prepare fluorinated graphene nanosheets. In a previous study, it was found that graphene was used as an ionic liquid [BMIM]BF_4_ additive, which showed good tribological performance under vacuum conditions. In order to further improve its performance, fluorinated graphene was selected as an additive to study the tribology of fluorinated graphene nanosheets in ionic liquids.

## 2. Materials and Methods

### 2.1. Preparation and Characterization of Fluorinated Graphene

Fluorinated graphite (Shanghai Fubang Chemical Co., Ltd., Shanghai, China, http://carfluor.company.lookchem.cn, accessed on 7 April 2023), N-methyl pyrrolidone (NMP), acetone, and ethanol of analytical grade were all purchased from Tianjin Chemical Reagent Co., Ltd. (Tianjin, China) and used directly. Further description of the main chemicals is included in Table 1.

FG nanosheets were prepared according to the method of liquid-phase ultrasonic exfoliation of graphite [43]: We weighed 10 g of fluorinated graphite, dispersed it in 100 mL of NMP solution, and ultrasonically dispersed it with a cell crusher for 10 h, power at 125 W, ultrasonic 2 s, stop 1 s. The pretreated graphite fluoride (about 10 g) was filtrated with a microporous membrane. We dispersed the pretreated graphite fluoride in a new 100 mL NMP solution and continued to sonicate for 24 h with the same ultrasonic power and time settings. We used a centrifuge to separate the graphite fluoride with larger particles and thicker lamellae by centrifugation at 500 rpm for 45 min. Then we increased the rotational speed to 8000 rpm, centrifuged for 10 min, and obtained FG nanosheets (FG-1) with larger lamellae; we continued to increase the rotational speed to 10,000 rpm and centrifuged for 10 min to obtain fluorinated graphene with smaller lamellae (FG-2). The obtained FG nanosheets were washed several times with acetone and ethanol. The morphology and structure of the fluorinated graphene after liquid-phase ultrasonic exfoliation were observed by Field Emission Scanning Electron Microscopy (FESEM, JSM-6701F, JEOL, Tokyo, Japan) and transmission electron microscopy (TEM, FEI Tecnai F300, Waltham, MA, USA). The microstructure of the prepared FG nanosheets was characterized by X-ray diffraction (XRD, Rigaku, D/max-2400, Tokyo, Japan) and a thermal analyzer (TGA-7, PerkinElmer, Waltham, MA, USA).

### 2.2. Dispersion of Fluorinated Graphene in IL and Preparation of a Composite Lubricating Film

First, a suspension of 2.5 mg/mL IL-FG (FG-1, FG-2 separately) was prepared by adding 10 mg of FG nanosheets to 4 mL of IL (1-butyl-3-methylimidazolium tetrafluoroborate [BMIM]BF_4_). Then, ultrasonic dispersion was carried out with a sonicator for 2 h. Then, 90 μL of the suspension was taken from 2.5 mg/mL IL-FG and diluted to 3 mL with IL, and ultrasonic dispersion was continued for 2 h. A homogeneous suspension of 0.075 mg/mL IL-FG was obtained. An appropriate amount of the liquid lubricant, as mentioned above, was taken and uniformly coated on the surface of the stainless steel substrate, and the thickness of the coating film was not more than 2 μm to obtain a solid–liquid composite lubricating film.

### 2.3. Testing Methods

IL-G and IL-FG friction behaviors were investigated on the vacuum friction tester developed by the Lanzhou Institute of Chemical Physics, Chinese Academy of Sciences. The test was carried out at room temperature, the vacuum degree was 10^−3^ Pa, and the friction pair was GCr15 balls with Φ3 mm. The load was 5 N, the rotation radius was 6 mm, the sliding speed was 40 rpm, and the test time was 120 min. A surface profiler (Alpha-Step D-100, Milpitas, CA, USA) was used to observe the cross-sectional morphology of the wear scar and measure the wear volume. Each test was repeated 3 times, and the average value was obtained. The wear rate was calculated by the formula [44]. The wear surfaces and transfer films were analyzed via an optical microscope (STM6, Olympus, Tokyo, Japan), Raman spectroscopy (Thermo Scientific DXR, Waltham, MA, USA, laser power of 4.4 mW, laser wavelength of 780 nm, and objective at MPlan 10×/0.25 BD), scanning electron microscopy (SEM, TESCAN VEGA3 SBU, SEM HV: 20.0 kV), and X-ray photoelectron spectroscopy (XPS, Thermo Fischer, Waltham, MA, USA, ESCALAB 250Xi). 

## 3. Results

### 3.1. Structural Analysis of Fluorinated Graphene

Figure 1 shows the morphology of graphite fluoride, in which Figure 1a is a TEM image. It can be seen that the graphite fluoride presents an apparent lamellar structure, and the lamellae are thicker and larger in size and block. The FESEM (Figure 1b) can see the multilayer structure of fluorinated graphene, and part of the fluorinated graphene nanosheets are exfoliated from the surface of fluorinated graphite. Selecting the NMP solution as the solvent, NMP molecules can intercalate between the graphite fluoride layers and adsorb to the surface of the lamellae through π–π interaction, resulting in the weakening of the van der Waals force between the graphite fluoride layers, which is more conducive to the exfoliation of the graphene fluoride layer [40,41]. Moreover, the NMP solution is miscible with water and soluble in ether, acetone, and other organic solvents, so the NMP molecules adsorbed on the sample surface are washed away by other solvents [42]. Figure 2 shows FG nanosheets obtained by liquid-phase ultrasonic exfoliation with different sheet sizes. The FG nanosheets become smaller, the thickness thinner, and the sheet’s surface flat. Figure 2a shows the FG-2 with a thinner sheet, which is easier to adsorb onto the copper substrate due to the thinner sheet forming a thin film.

When X-rays are irradiated on an object, coherent scattering occurs, and the coherent scattered waves of each atom may interfere with each other in a specific direction in space and cause diffraction. The diffraction lines’ direction, intensity, and line shape contain much information on the structure of matter. XRD analysis is a material structure analysis method based on X-ray diffraction. Here we used XRD to accurately measure the 2θ angle data of the diffraction peak position. Figure 3 shows the X-ray diffraction patterns of FG and FG nanosheets. It can be seen from the figure that the 2θ = 14° diffraction peak corresponds to the (001) plane, which indicates that the insertion of fluorine atoms increases the interlayer spacing of graphite atoms [43], which significantly weakens the interaction between layers and also indicates that the fluorine content of fluorinated graphite is higher. Compared with fluorinated graphite, FG nanosheets have a new broad and weak diffraction peak at 2θ = 21° (002), which reflects the increased interplanar spacing of FG nanosheets. The peaks at 41° and 73° in the figure correspond to the (10) plane and the (11) plane, respectively [44], which reflect the length of the C–C bond in the hexagonal mesh plane. It can be seen from the figure that the smaller the sheet of FG nanosheets, the smaller the two peaks.

Figure 4 shows the thermal analysis curves of fluorinated graphite and FG nanosheets. The test conditions are heating temperature 50–800 °C, under N_2_ atmosphere, heating rate 20 °C/min. From the thermograms, it is found that both fluorinated graphite and FG nanosheets are very stable before 350 °C with almost no mass loss. The thermal stability of fluorinated graphite is higher than that of fluorinated graphene above 350 °C due to the different surface energy. Graphite fluoride is a multilayer structure; most of the C–C and C–F bonds are wrapped inside the sheet, and only a small part of the C–C and C–F bonds are exposed to the outside. While FG nanosheets are thinner, more C–F bonds are exposed, which is unstable and prone to decomposition after heating. Thus, the thermal stability of FG nanosheets is poor [40].

### 3.2. Thermal Stability of IL-GO, IL-FG-1, and IL-FG-2

The temperature range of the space field is about minus 120 to 150 °C, and the space lubricant needs good thermal stability. As shown in Figure 5, three kinds of nanofluids decompose at close to 400 °C. Combined with the thermal analysis diagram in Figure 4, it can be understood that GO and FG nanosheets are more stable and begin to decompose at close to 500 °C. Thus, in Figure 5, it is the ionic liquid that is breaking down around 400 °C. This is consistent with the previous thermogravimetric analysis results [10]. According to literature reports, the products obtained from IL decomposition are mainly boron trifluoride, hydrogen fluoride, methyl fluoride, and 1-butene [45].

The thermal analysis test results show that adding additives has little effect on the thermal stability of the ionic liquid, and the quality below 300 °C does not decrease and remains stable. It shows that the nanofluid has good thermal stability. It can meet the conditions of use in space. 

### 3.3. Tribological Properties

Figure 6 shows the curve relationship of the friction coefficient of ionic liquids with different additives as a function of time. It can be seen from the figure that the friction coefficient of the initial stage of friction is relatively high and unstable. After a few minutes, the friction coefficient of the three nanofluids tends to be stable. After about 67 min, the friction coefficient of IL-G increases significantly and fluctuates greatly. The friction coefficients of the other two nanofluids remained stable. Wear rate and friction coefficient have the same trend of change. Figure 7 shows a photo of the surface topography of the steel sheet and steel ball worn under an optical microscope after ultrasonic cleaning with acetone in an ultrasonic instrument. It can be seen from the comparative analysis that when graphene is added to the ionic liquid, after a long-term friction test, there are not only many scratches or furrows on the surface of the wear spot, but they are also thick and deep. However, after adding FG nanosheets, the scratches and furrows become fine and shallow, and the surface becomes smooth. The wear scar width of IL-FG-2 is smaller than that of IL-FG-1. It shows that the smaller and thinner FG nanosheets are easier to adsorb onto the surface of the friction pair to form a protective layer, and it is easier to enter the microscopic defect area of the worn surface to fill the defects.

To further investigate the improvement of FG nanosheets on the antiwear performance, the surface of the wear spots was observed by Raman and SEM, and the results are shown in Figure 8 and Figure 9. The D-band at ~1357 cm^−1^ and G-band at ~1591 cm^−1^ are ascribed to carbon additives [46]. In Figure 8, the corresponding Raman spectra of the wear scar on the steel plate confirm the presence of nanocarbon film and deposit on the steel. In Figure 8a, the black line is the test results of Raman spectrum on the steel wear scar, because very little GO deposited on wear scar of the steel, the Raman spectrum in Figure 8a indicates the steel wear scar is very low. Therefore, the corresponding red curve was obtained by fitting the Raman spectrum results.

From Figure 9a,b, it can be seen that when graphene was used as lubricant filler, the scratches on the wear surface are more obvious, and there are black spots at the depressions of the wear spots. Figure 9 also shows the energy spectrum test conducted at the depressions of the wear spots, the data is shown in Table 2. The surface contains C, N, F, Si, Cr, Fe, and Ni, among which Si, Cr, Fe, and Ni are mainly from the steel sheet; N and F are from IL; and C is mainly from graphene. When graphene fluoride is used as a lubricant filler, the abrasion marks become fine and shallow, and the abrasion chips can be observed to fill in the depressions of the abrasion marks. The composition of the surface of the abrasion marks can be tested by EDS spectroscopy (Figure 9), which shows that the composition is the same as that of IL-G. However, the content of F is increased, so it can be presumed that F comes from FG nanosheets, which proves that graphene fluoride is adsorbed on the contact surface of the friction substrate and forms a protective film during the friction process. 

X-ray photoelectron spectroscopy is a powerful technique providing information about elemental composition and binding conditions. As shown in Figure 10a, four peaks appeared in the C 1s XPS spectrum of wear scar. The peak at 284.8 eV is assigned to the CH_2_ group in the alkyl chain of ILs, whereas the peak at 286.4 eV can be assigned to the C atoms bonded to the N atoms (C*-N in the imidazolium ring) [10]. Two additional C-F peaks are found between 288.0 and 292.0 eV after using IL-FG-2 because of the presence of FG nanosheet film on the wear scar of the steel plate after friction testing. Figure 10b shows that the prominent F peak at 688.8 eV after friction testing is additional spectroscopic evidence of C-F bonds. There are corresponding peaks in the Fe 2p spectrum (Figure 10c), and the peaks of Fe at 706.5 eV and 709.2~712.0 eV belong to FeF_2_ or FeF_3_, respectively [9,10]. When GO is used as an additive, F and Fe in ionic liquids react to form FeF_2_ or FeF_3_, and the FG nanosheet contains more F, so more FeF_2_ or FeF_3_ is formed.

According to XPS data, it can be inferred that the process of GO and FG nanosheets producing lubrication can be regarded as a process of mechanical and chemical interaction. In the early stage of the tribological test, the thin GO and FG nanosheets easily adsorb onto the metal surface, forming a deposition film on the metal surface to fill and isolate, thereby reducing the direct contact of the surface and reducing the friction of the sliding surface [47]. In the friction process, F^-^ in IL and FG nanosheets and the metal substrate surface chemically react, forming FeF_2_ or FeF_3_ film. Film containing the F-Fe chemical reaction has excellent lubricating properties and can effectively prevent direct contact between metal surfaces. FG nanosheets that contain more F in the friction process form more FeF_2_ or FeF_3_, so compared with GO additives, FG nanosheet additives can further reduce the wear.

### 3.4. Related Wear Mechanisms

Figure 11 shows a schematic of the mechanisms of the ILs with GO or FG nanosheets during friction. The early formation of a GO or FG nanosheet transfer layer avoids damaging two mating wear surfaces. Thus, the friction coefficient is low at the early stage. However, the GO nanosheets become significantly thicker due to overlapping after long friction testing [10]. In the friction process, F^−^ from ILs reacts with Fe in the steel sheet to create more F compound, which can also protect the steel sheet. Moreover, FG nanosheets contain more F^−^, so more Fe-F compound is formed on the steel surface, which can protect the steel surface and reduce friction and wear.

## 4. Conclusions

Using fluorinated graphite as the raw material, fluorinated graphene was successfully prepared by liquid-phase ultrasonic exfoliation. This preparation method is simple, and fluorinated graphene of different sheet sizes can be obtained under different centrifugal separation speeds.The tribological properties of FG nanosheets were studied. Compared with IL-G, IL-FG nanosheets have better tribological properties for steel/steel friction pairs; during the friction process, FG nanosheets are adsorbed on the interface of steel/steel friction pairs to form a protective layer to avoid friction. During the friction process, the FG-2 nanosheets with smaller lamellae can be deposited on the microscopic defect area of the worn surface under the action of compressive stress, thus playing a good role in repairing the worn surface.

## Figures and Tables

**Figure 1 materials-16-03929-f001:**
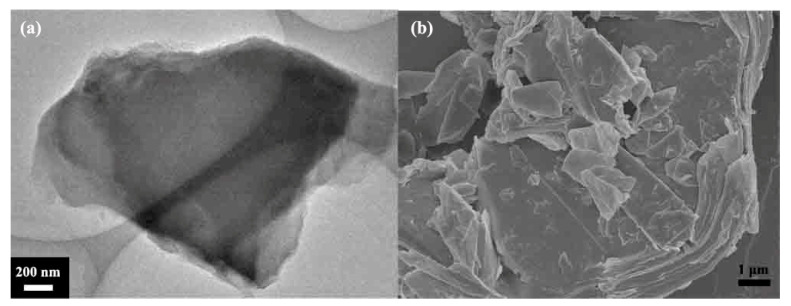
(**a**) TEM image of fluorinated graphite, (**b**) FESEM image of fluorinated graphite.

**Figure 2 materials-16-03929-f002:**
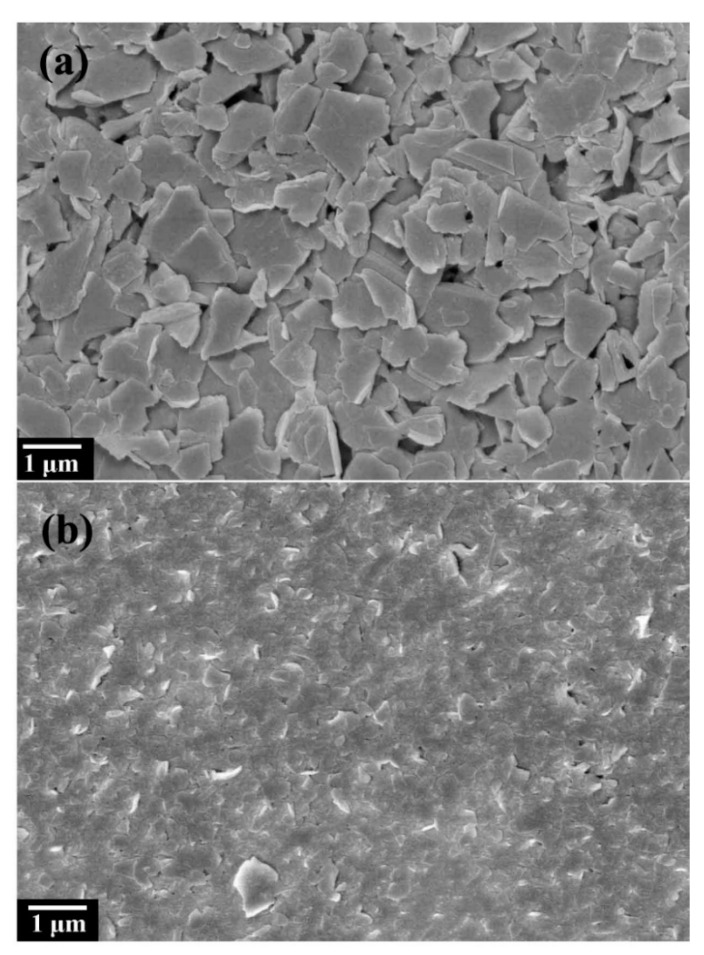
FESEM image of FG nanosheets-1 and FG nanosheets-2, (**a**) FG nanosheets-1, (**b**) FG nanosheets-2.

**Figure 3 materials-16-03929-f003:**
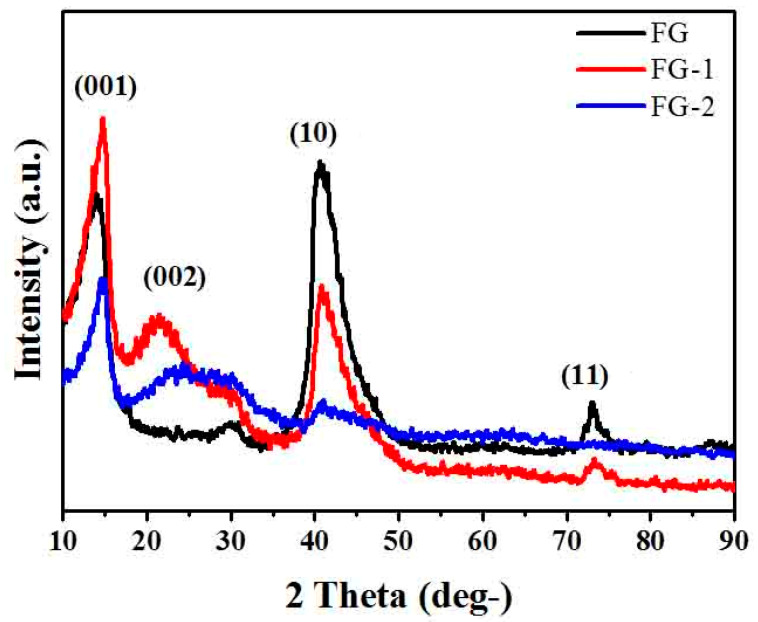
XRD patterns of fluorinated graphite and fluorographene.

**Figure 4 materials-16-03929-f004:**
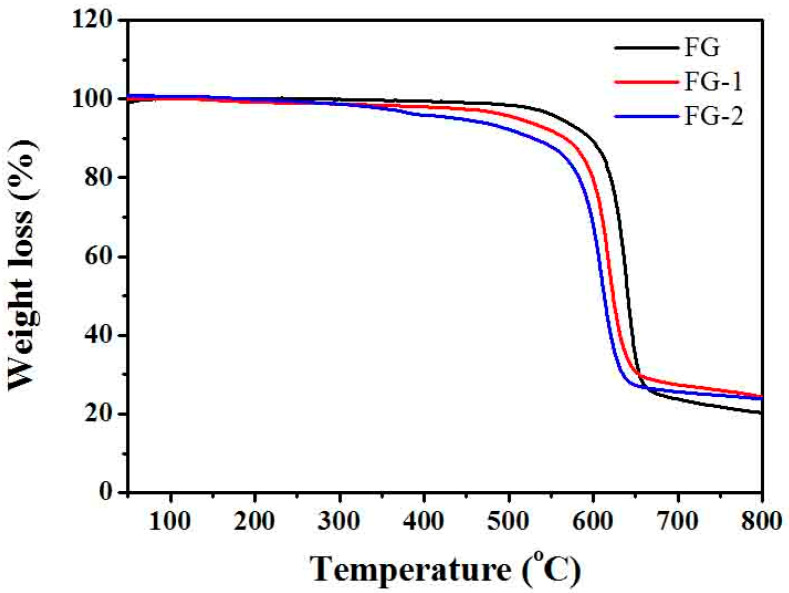
TGA curves of fluorinated graphite and fluorographene.

**Figure 5 materials-16-03929-f005:**
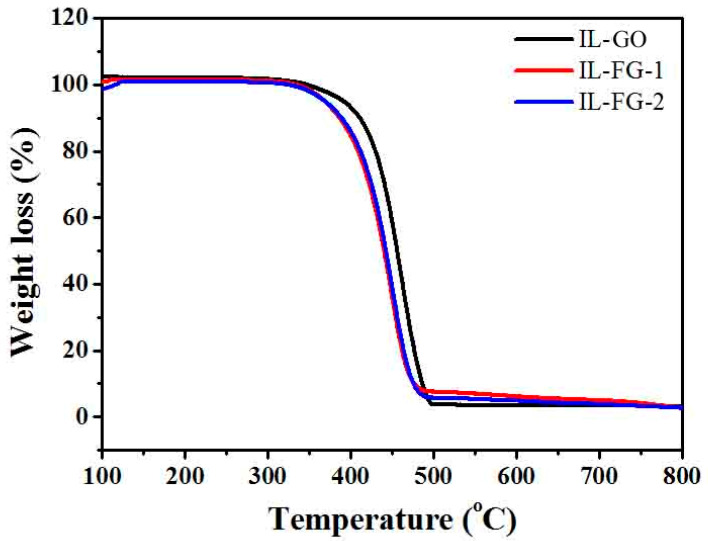
TG profiles of IL-G, IL-FG-1, and IL-FG-2.

**Figure 6 materials-16-03929-f006:**
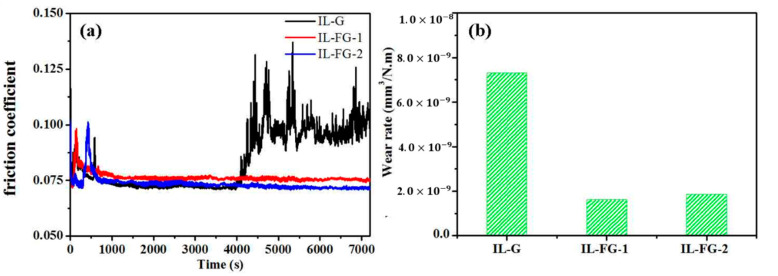
Friction coefficient and wear rate with lubrication of IL-G, IL-FG-1, and IL-FG-2, (**a**) Friction coefficient, (**b**) wear rate.

**Figure 7 materials-16-03929-f007:**
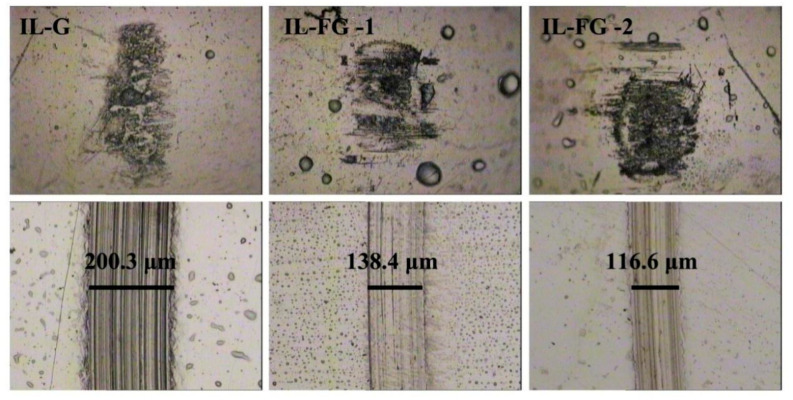
Optical micrograph images of the steel plate and steel ball wear scar.

**Figure 8 materials-16-03929-f008:**
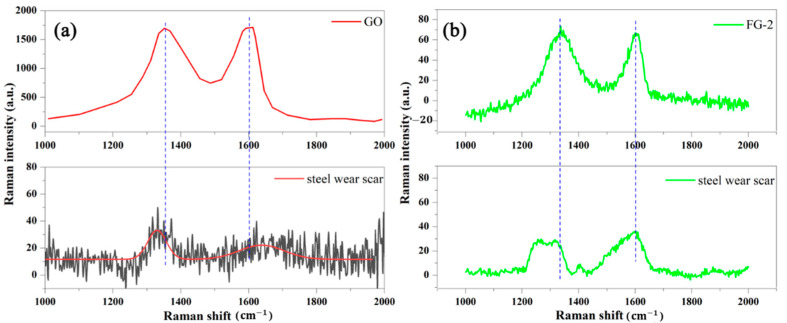
Raman spectra of (**a**) pristine GO and wear scar on the steel plate, (**b**) pristine FG-2 and wear scar on the steel plate.

**Figure 9 materials-16-03929-f009:**
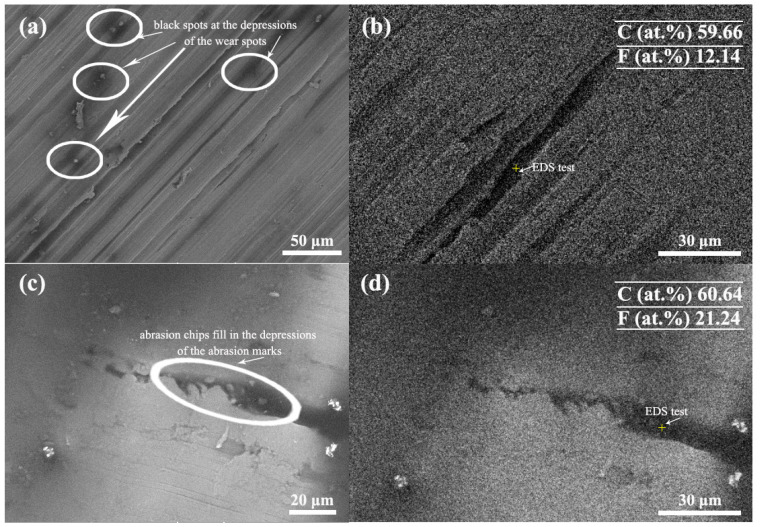
SEM images of wear scar under the same load and, respectively, by IL and IL-FG-2. (**a**) wear scar of IL-GO, (**b**) EDS test at the depressions of the wear spots after using IL-GO. (**c**) wear scar of IL-FG-2, (**d**) EDS test at the depressions of the wear spots after using IL-FG-2.

**Figure 10 materials-16-03929-f010:**
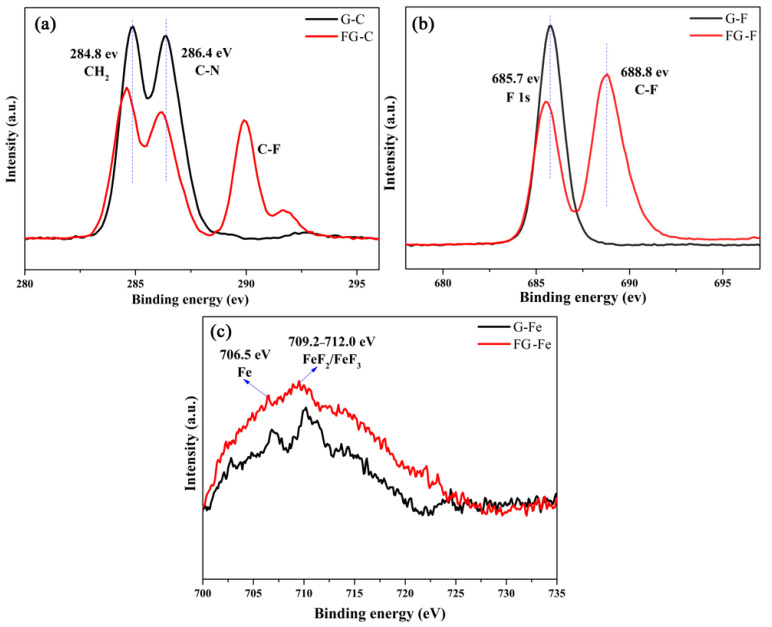
XPS spectra of wear scar of steel plate after friction testing: (**a**) C 1s spectra; (**b**) F 1s spectra; (**c**) Fe 2p spectra.

**Figure 11 materials-16-03929-f011:**
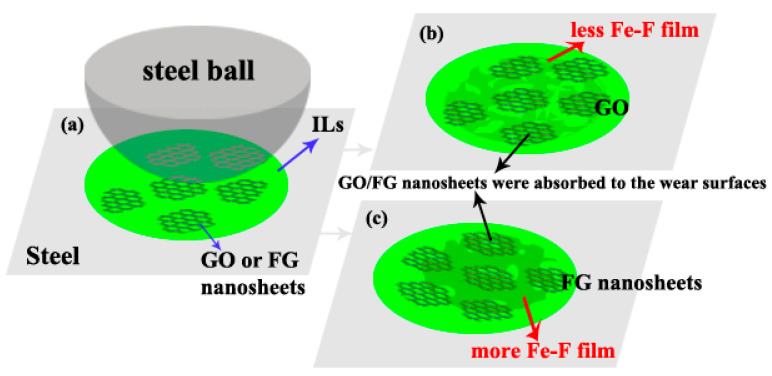
Schematic presentation of friction mechanism using different additives, (**a**) schematic presentation of friction using GO or FG nano-additives, (**b**) GO nanosheets were absorbed to the wear surfaces, (**c**) FG nanosheets were absorbed to the wear surfaces.

**Table 1 materials-16-03929-t001:** Technical datasheet of main materials in this experiment.

Materials	Manufacturer	Purity
ionic liquids (IL [BMIM]BF4)	State Key Laboratory of Solid Lubrication, Lanzhou Institute of Chemical Physics (Lanzhou, China)	97%
graphene oxide (GO)	Nanjing XFNANO Materials Tech Co., Ltd. (Nanjing, China)	~99%
fluorinated graphite (FG)	Shanghai Fubang Chemical Co., Ltd., http://carfluor.company.lookchem.cn	99.99%
N-methyl pyrrolidone (NMP)	Tianjin Chemical Reagent Co., Ltd.	analytical grade 99.7%
acetone	Tianjin Chemical Reagent Co., Ltd.	analytical grade 99.7%
ethanol	Tianjin Chemical Reagent Co., Ltd.	analytical grade 99.7%

**Table 2 materials-16-03929-t002:** The surface compositions (at.%) of steel after friction test.

	C (at.%)	N (at.%)	F (at.%)	Si (at.%)	Cr (at.%)	Fe (at.%)	Ni (at.%)
IL-G	59.66	11.14	12.14	0.11	3.34	12.40	1.21
IL-FG	60.04	13.63	21.41	0.04	3.47	12.49	1.03

## Data Availability

The data used to support the findings of this study are available from the corresponding author upon request.

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
