# Peer review of "The Preparations of Fluorographene Nanosheets and Research in Tribological Properties in High Vacuum"

_materials, 2023, doi:10.3390/ma16113929_

Round 1
Reviewer 1 Report
This paper could be suitable for publication after adding much technical information.
1/ It seems that the authors haven't checked the use of indices and exponents so the text is not pleasant to read (everywhere, lines 51,64,65,68,70,168, ...)
2/From the given reference line 135 (Shanghai Fubang Chemical Co., Ltd), it's impossible to find the supplier online, so please add a link presently available in the reference.
Moreover, I assume that the technical datasheet is available to extract more specific information, complementary to the values provided in the paper or which could be compared with.
3/ In subparagraph 2.3, what are the conditions for all the cited experiments?
For Raman spectroscopy, give the laser power, the wavelength (as the shape of Carbon material is sensitive to the wavelength), and the objective.
For SEM, what is the working voltage?
4/ Fig 3: XRD is typical of turbostratic stacking, as the 10 and 11 peaks are asymmetric to high 2theta values. In such a situation, the index is not 100 but 10 as there is no 3D unit cell.
It's really old physics (see for example: Biscoe, J., & Warren, B. E. (1942). An x‐ray study of carbon black. Journal of Applied Physics, 13(6), 364-371.)
5/ For the X-Ray and for Raman, is there a background removal? it's always important as what is considered background is a signal and could sometimes be used to go further in the interpretation.
6/ In Raman's journals, (a.u.) is prohibited as it's the astronomic unit and is replaced by (arb. u.) for the arbitrary unit. Correct the ordinates in the figures.
Author Response
Reply to the reviewers’ comments and revision details
Title: The Preparations of Fluorographene nanosheets and Research in Tribological Properties in High Vacuum
Dear editor and reviewers,
Thanks a lot for your patient and constructive comments. We have carefully revised our manuscript by considering the reviewer’s comments. A point-by-point reply to the reviewer’s comments and revision details are as follows (in order of the reviewer’s comments):
Reviewer 1
This paper could be suitable for publication after adding much technical information.
(1) It seems that the authors haven't checked the use of indices and exponents so the text is not pleasant to read (everywhere, lines 51,64,65,68,70,168, ...)
Response:
We have carefully corrected the indices and exponents for all the revised manuscript. The modified sections are highlighted in the revised manuscript.
(2) From the given reference line 135 (Shanghai Fubang Chemical Co., Ltd), it's impossible to find the supplier online, so please add a link presently available in the reference. Moreover, I assume that the technical datasheet is available to extract more specific information, complementary to the values provided in the paper or which could be compared with.
Response:
The link of supplier fluorinate graphite has been added in the reference. The relevant technical datasheet of materials in this experiment was also added. The modified sections are highlighted in the revised manuscript, page 3-4.
Table 1 Name, manufacturer and purity of main materials in this experiment
|
Name |
Manufacturer |
Purity |
|
Ionic liquids (IL [BMIM]BF4) |
State Key Laboratory of Solid Lubrication, Lanzhou Institute of Chemical Physics |
97% |
|
Graphene oxide (GO) |
Nanjing XFNANO Materials Tech Co., Ltd. |
~99% |
|
Fluorinated graphite(FG) |
Shanghai Fubang Chemical Co., Ltd. http://carfluor.company.lookchem.cn |
99.99% |
|
N-methyl pyrrolidone (NMP) |
Tianjin Chemical Reagent Co., Ltd. |
analytical grade 99.7% |
|
acetone |
Tianjin Chemical Reagent Co., Ltd. |
analytical grade 99.7% |
|
ethanol |
Tianjin Chemical Reagent Co., Ltd. |
analytical grade 99.7% |
(3) In subparagraph 2.3, what are the conditions for all the cited experiments?
For Raman spectroscopy, give the laser power, the wavelength (as the shape of Carbon material is sensitive to the wavelength), and the objective.
For SEM, what is the working voltage?
Response:
For Raman spectroscopy, laser power is 4.4 mW, laser wavelength is 780 nm, and the objective is MPlan 10×/0.25 BD.
Working voltage of SEM is 20.0 kV. The modified sections are highlighted in the revised manuscript, page 5.
(4) Fig 3: XRD is typical of turbostratic stacking, as the 10 and 11 peaks are asymmetric to high 2theta values. In such a situation, the index is not 100 but 10 as there is no 3D unit cell.
It's really old physics (see for example: Biscoe, J., & Warren, B. E. (1942). An x‐ray study of carbon black. Journal of Applied Physics, 13(6), 364-371.)
Response:
Thank you for your suggestion. Through the study of relevant literature, I learned more about the XRD analysis of carbon, and modified it in the article and diagram. The modified sections are highlighted in the revised manuscript, page 6-7.
Figure 3. XRD patterns of fluorinated graphite and fluorographene.
(5) For the X-Ray and for Raman, is there a background removal? it's always important as what is considered background is a signal and could sometimes be used to go further in the interpretation.
Response:
Background has been removed for Raman test. X-Ray did not removal the background.
(6) In Raman's journals, (a.u.) is prohibited as it's the astronomic unit and is replaced by (arb. u.) for the arbitrary unit. Correct the ordinates in the figures.
Response:
Thank you for your suggestion. The ordinates have been corrected in the Figure 8.
Figure 8. Raman spectra of (a) pristine GO and wear scar on the steel plate, (b) pristine FG-2 and wear scar on the steel plate.

Reviewer 2 Report
The manuscript "The Preparations of Fluorographene nanosheets and Research in Tribological Properties in High Vacuum", written by Lili et al., describes an interesting approach for to use of fluorographene in a combination with ionic liquids in tribological applications. The manuscript is well written, data described and discussed. The only main issue I identified is the lack of a description of the graphene used for a comparison studies. I believe it would be the same material used in the authors previous paper. However, it would be ideal to include a description of characteristics, micrographs and Raman spectra of used graphene also in this manuscript. I have also few minor comments:
1. Thermal stability of the material(s) should be described better. What is the interpretation of the weight loss at cca 400˚C? What is the structure of the remaining material?
2. The quality of the Raman spectrum in the figure 8(a) - steel wear scar is extremely low. I suggest to measure the sample again using better parameters.
Author Response
Reply to the reviewers’ comments and revision details
Title: The Preparations of Fluorographene nanosheets and Research in Tribological Properties in High Vacuum
Dear editor and reviewers,
Thanks a lot for your patient and constructive comments. We have carefully revised our manuscript by considering the reviewer’s comments. A point-by-point reply to the reviewer’s comments and revision details are as follows (in order of the reviewer’s comments):
Reviewer 2
The manuscript "The Preparations of Fluorographene nanosheets and Research in Tribological Properties in High Vacuum", written by Lili et al., describes an interesting approach for to use of fluorographene in a combination with ionic liquids in tribological applications. The manuscript is well written, data described and discussed. The only main issue I identified is the lack of a description of the graphene used for a comparison studies. I believe it would be the same material used in the authors previous paper. However, it would be ideal to include a description of characteristics, micrographs and Raman spectra of used graphene also in this manuscript.
Response:
Thank you for your suggestion. GO well dispersed in IL([BMIM]BF4) than graphene, so GO was choosed in this research. Graphene really should be used as a comparison studies. I will consider it in the next step of research.
I have also few minor comments:
(1) Thermal stability of the material(s) should be described better. What is the interpretation of the weight loss at cca 400˚C? What is the structure of the remaining material?
Response:
As shown in figure 5, three kinds of nanofluids decompose at close to 400 oC. Combined with the thermal analysis diagram in Figure 4, GO and FG nanosheets are more stable and begin to decompose at close to 500 oC. So in Figure 5, it's the ionic liquid that's breaking down around 400 oC. This is consistent with the previous thermogravimetric analysis results [10]. According to literature reports, the products obtained from IL decomposition are mainly boron trifluoride, hydrogen fluoride, methyl fluoride and 1-butene[46]. The modified sections are highlighted in the revised manuscript, page 8.
[46] Zhang L.Y., Liu S.H., Wang Y., Exploring the influence of the type of anion in imidazolium ionic liquids on its thermal stability, Journal of Thermal Analysis and Calorimetry, 2023.
Figure 5. TG profiles of IL-G, IL-FG-1, and IL-FG-2.
Previous thermogravimetric analysis results in reference 10.
(2) The quality of the Raman spectrum in the figure 8(a) - steel wear scar is extremely low. I suggest to measure the sample again using better parameters.
Response:
Thank you for your suggestion. The sample was measured again, the new test results shows as follows. Because very little GO deposited on wear scar of the steel, the Raman spectrum in the figure 8(a) - steel wear scar is still low.
Figure 8. Raman spectra of (a) pristine GO and wear scar on the steel plate, (b) pristine FG-2 and wear scar on the steel plate.

Round 2
Reviewer 1 Report
The paper has been corrected and is now suitable for publication.
Reviewer 2 Report
Authors considerably improved the quality of the manuscript and adequately answered all my questions and comments. I thus recommend this manuscript for publication.